# Effectiveness of Guided Breathing and Social Support for the Reduction of Pre-Exam Anxiety in University Students: A Factorial Study

**DOI:** 10.3390/healthcare11040574

**Published:** 2023-02-15

**Authors:** Lucía Ortega-Donaire, Cristina Álvarez-García, María Dolores López-Franco, Sebastián Sanz-Martos

**Affiliations:** Department of Nursing, Faculty of Health Sciences, University of Jaén, 23071 Jaén, Spain

**Keywords:** anxiety, university students, relaxation therapy

## Abstract

Anxiety is a state of mind that university students often manifest in exam situations, which can negatively impact their grades. This study aimed to evaluate the effect of different relaxation techniques, including guided breathing and social support, on test anxiety among nursing students a few minutes before taking the final knowledge assessment test. For this purpose, a factorial study with a post-intervention measurement was carried out with three groups of nursing students. One group used the full yogic breathing relaxation technique (abdominal, thoracic, and clavicular), another used a social support technique, and the last group did not receive any intervention. Of 119 participants, 98.2% showed a moderate-high level of anxiety. Regarding the anxiety scale score, it was found that participants with moderate anxiety levels had higher scores on the knowledge test (Rho = −0.222; *p* = 0.015). The present study found no differences in anxiety levels between the study groups. Combining these relaxation techniques with others shown to be effective could reinforce their positive effect. Starting to work on this anxiety from the beginning of nursing courses appears to be a good strategy, striving to improve students’ confidence in their abilities.

## 1. Introduction

Test anxiety is defined as a situational-specific trait characterized by the presence of restlessness, intense excitement, distress, or a feeling of insecurity associated with test-taking environments [1]. This unpleasant feeling prior to an assessment test is largely due to the great social pressure students are subjected to and the intense focus on achieving high academic performance [2].

This situation presents differential characteristics to the general feeling of anxiety, presenting a greater phobic sensation [3]. Associated with this feeling of phobic anxiety is a situation perceived as a danger due to the response of the sympathetic nervous system through the activation of body systems.This diminishes the ability to concentrate, leading to a loss of ability to show the skills or knowledge acquired during the assessment process. Faced with these situations, we find ourselves with a double problem. On the one hand, the objective test loses reliability, which is not associated with a lack of acquisition of the specific construct of study but with a reduction in the ability to develop it on the part of the students. On the other hand, the body’s systems respond in an amplified way to this feeling of anxiety during future assessments, reinforced by the previous unpleasant real experience [4,5]. 

The feeling of anxiety before and during tests generates consequences in students, which makes it necessary to develop measures to reduce it, allowing for increased mastery over anxiety, control of the situation, and increased self-esteem [6]. When considering new strategies for coping and managing test anxiety, a number of agents can act as channels of contact with students, such as classmates through group work, parents, academic advisors, and of course, the teachers involved in their training [7]. In addition, teachers should be aware of the need to develop measures to deal with test anxiety and know which measures are most effective in dealing with each situation or provide various tools, seeking to develop skills and competencies of control and management in students [8].

A previous multicenter study conducted in Spain with a sample of 28,559 students from 16 Spanish universities found that 20.84% reported significant anxiety when facing exams and, in some cases, needed specialized help [2]. Furthermore, previous research on university students in healthcare degree programs has found a prevalence of pre-exam anxiety in 30–50% of students, which is higher than in other university degrees. This difference with respect to other university students indicates the need to address the group of university students in general due to the consequences of anxiety before exams, specifically in healthcare degrees due to their higher prevalence [4,9].

A relaxation technique is a method that teaches certain forms of behavior through one’s own body to reduce the physiological activation of the organism. In literature, different interventions are described regarding the management of test anxiety, differentiating between those that are carried out during the exams (intra-test) or those that are carried out outside the classroom as strategies to prepare students to take these tests (pre-test) [8].

Regarding the intra-test intervention, there are different treatment modalities to reduce this anxiety. Among them is systematic desensitization, which consists of a hierarchical implementation of stressful stimuli through images, which, compared with the technique of cognitive restructuring or cognitive therapy in 50 Turkish students in their final year of studies, found a decrease in anxiety in both groups [10]. Jacobson’s progressive relaxation is another technique that can be used during the assessment test. It consists of tensing and releasing a series of muscles to the rhythm of slow and calm breathing, perceiving the body sensations produced, and thus almost completely eliminating muscular tensions and contractions, inducing deep relaxation [11]. One hundred fourteen medical students from South Gujarat University performed this technique during their university exams. They found reduced anxiety and stress between their values before and after the intervention [12]. Finally, guided breathing and direct contact with university students have reported significant benefits in reducing test anxiety, as reported by several previous studies [13,14,15,16,17].

The second modality of pre-test interventions differs in terms of the time elapsed between the application of the intervention and the performance of the skills or knowledge test, with a period of time for the internalization of the content addressed in the evaluations. Although, we could find ourselves with a reduction in effectiveness by requiring the relaxation guidelines to be correctly applied at the time of the peak of the stressful situation. Furlan et al. [18] implemented an intervention program of 12 two-hour weekly meetings with medical and psychology undergraduates with high test anxiety. This type of intervention produced favorable results in those participants who completed it, experiencing improvements and finding a healthier coping strategy for the evaluation test. Another intervention modality, which combines music therapy and aromatherapy 20 min before the assessment test, found reduced anxiety and stress levels and improved performance of fundamental nursing skills [19]. 

Systematic reviews indicate the need to verify the efficacy of interventions both from studies that have already been conducted and from other interventions of unproven efficacy [8]. Therefore, this study aimed to determine the effect of guided breathing and social support on test anxiety among nursing students when administered minutes before a final knowledge assessment test. As a research question, we asked whether guided breathing or social support effectively reduces anxiety when used before an evaluation test, comparing anxiety levels after these two techniques with no intervention. The choice of these specific techniques was due to prior knowledge of them by the students; they do not excessively distract the students and do not alter the time available to take the test. In addition, the ease of performing both techniques, the few tools necessary to carry them out, and the positive effects found in other research were decisive in their choice [13,14,15,16,17].

## 2. Materials and Methods

### 2.1. Design

The study was designed as a factorial study with a post-intervention measure. This study shows the differences between the two intervention and control groups in the level of anxiety and its relationship to the knowledge score obtained in the test.

### 2.2. Sample and Setting

The participants were nursing students from the third year of the Nursing degree at the University of Jaen during the 2021–2022 academic year, who voluntarily agreed to participate by signing the informed consent form. A few days before the test, it was explained that this experiment would be carried out, and it finally took place in May 2022. We included all participants who voluntarily agreed to participate. Participation or non-participation in this study did not in any way affect the given grade in any course, nor did it lead to any reward. Participants taking any medication that could affect their anxiety levels were excluded. Based on previous research [3,20], a sample size calculation was established to detect a difference of 5.0 points in the anxiety level, with a standard deviation of 9.5 points, a confidence level of 95%, and a power of 80%. Therefore, the minimum sample size was established at 116 participants.

### 2.3. Data Collection

The study was carried out using 3 randomized groups of participants. Assignment of participants to the evaluation groups was performed using the list of students for random selection by clusters in forming the three groups. The assignment process was carried out with Epidat version 4.2 for windows. The interventions that were carried out were only explained prior to their implementation.

One group used the full yogic breathing relaxation technique (abdominal, thoracic, and clavicular) and was guided at all times by the classroom teacher. The intervention began with an introduction, teaching the students to control their breathing and to be aware of the breathing movements they were executing. Then, it was explained to the students how they should feel the air entering and leaving the body, with the air entering on inspiration through the nostrils, going on to feel it in the abdomen, retaining it in the thorax, and expelling it through the mouth, feeling the clavicular movement at the time of 3″ -2″ -1″, respectively.

The second group received a relaxation technique of rapprochement and contact with the students. This second intervention consisted of maintaining contact with the group of students present in the classroom, talking to them, and letting them express how they feel as a tool for eliminating accumulated tension. The last group did not receive any intervention. The groups were separated into different rooms 30 min before the test start time. The duration of the interventions was 15 min, after which the evaluation test was started. The questionnaire was completed only once after the intervention and before the evaluation test. The questionnaire, consisting of two sections, was used to collect the following information (Appendix A): Sociodemographic data: Age and sex;Assessment of the level of anxiety: The “German Test Anxiety Inventory Argentinean Adaptation” scale translated into Spanish [21] was used. The time needed to answer the questionnaire was approximately 5 min. This questionnaire consisted of a Likert scale with 29 items, where 1 is extremely rare and 4 is most common. This questionnaire consists of 4 dimensions related to test anxiety: lack of confidence, worry, emotionality, and interference. The overall reliability of the scale and subscales was above Cronbach’s alpha value of 0.8, which is considered acceptable. The range of scores was between 29 and 116. The original scale adapted by Heredia et al. [21] was composed of 4 factors, which were maintained in the later revision created by Sesé et al. [22] and used among the Spanish population: Emotionality (8 items), worry (10 items), interference (5 items), lack of confidence (6 items). The anxiety scale score was recorded, with scores above 70% of the maximum score being considered high anxiety, between 50 and 70% moderate anxiety, and below 50% being considered low anxiety;Knowledge test score: This was assessed by the score obtained in the course knowledge test. The test consisted of 10 multiple-choice questions and 5 open-ended questions. The test correction was carried out by the course coordinator, who was blinded to the group to which the participants had been assigned. The scores for the knowledge test ranged from 0 to 10 points.

### 2.4. Statistical Analysis

A descriptive analysis of the qualitative variables was carried out, obtaining their frequency and percentage distribution. Measures of central tendency and dispersion were calculated for the quantitative variables. The normal distribution of the variables was tested with the Kolmogorov-Smirnov test. A student t-test and the Analysis of Variance test (ANOVA contrast with Fisher’s F) were used to assess the statistical significance of group differences concerning anxiety. In the case of non-parametric variables, the Mann-Whitney and the Kruskal-Wallis tests were used for independent variables with two or more categories, respectively. To study the influence of sex on the level of anxiety, a student *t*-test was used, and to study the correlation between age and level of knowledge, Pearson’s R was used. When variables were non-parametric, the Mann-Whitney test and Spearman’s Rho were used. For all analyses, a significance level of 95% was established. The analyses were performed with the statistical package SPSS version 27 for Windows. 

### 2.5. Ethical Considerations

The research ethics committee of the University of Jaen (Spain) approved the study protocol (JUL.22/3.PRY). The participants voluntarily signed an informed consent form and received written information on the study. Furthermore, the researchers ensured the confidentiality of the data collected. The participants were informed that their participation would not affect their course grades. The researchers are professors at the University of Jaen, but not in the course which was evaluated. No personal data were requested.

## 3. Results

We selected a group of 121 third-year nursing students, 119 of whom agreed to participate and met the inclusion criteria, giving a response rate of 98%. 92 (77.3%) were female, with a mean age of 22.70 (standard deviation (SD): 4.62) years. 

The anxiety scale showed a high level of reliability with a Cronbach’s alpha value of 0.84. The subscales showed acceptable reliability values with a range between 0.84 and 0.92. The mean score on the anxiety scale was 80.4 (SD: 10.7) points. The worry subscale showed the highest score before the assessment test. Table 1 shows the main characteristics of the sample.

Before the test, one participant showed a low level of anxiety, while 49.6% showed a high value, the same proportion as the moderate anxiety score.

As a result of the bivariate analysis, no statistically significant differences were found in the anxiety scale score in the intervention groups (*p* = 0.451). Regarding the anxiety scale score, we found a negative, significant weak correlation with the knowledge assessment test score, showing that participants with lower anxiety levels obtained higher scores on the knowledge test (Rho = −0.222; *p* = 0.015). We found significant weak-moderate correlations with the knowledge test score for the subscales of emotionality (Rho = −0.272; *p* = 0.003), worry (Rho = −0.183; *p* = 0.047), lack of confidence (Rho = 0.314; *p* = 0.001) and interference (Rho = −0.384; *p* < 0.001). For the sociodemographic variables, it was found that male participants presented significantly higher values in the lack of confidence subscale (Z = −2.532; *p* = 0.011). Analyzing the study groups, we found for the social support group, a statistically significant moderate negative correlation between the score on the knowledge scale and the subscale of emotionality (−0.389; *p* = 0.019), worry (−0.359; *p* = 0.029) and interference (−0.388; *p* = 0.018). For the guided breathing group and the control group, statistical significance was reached only for the subscale of lack of confidence (0.326; *p* = 0.037) in the first group and for interference (−0.350; *p* = 0.025) in the second group. Table 2 shows the results of all the hypothesis contrasts at the bivariate level.

## 4. Discussion

This study aimed to determine the effect of different relaxation techniques, including guided breathing and social support, on test anxiety among nursing students. The present study found that most nursing students presented moderate-high anxiety. The percentage of undergraduate nursing students with moderate-high anxiety varies in the literature, ranging from 75 [7] to 40% [23], supporting the need to study this phenomenon and try possible interventions to treat it. 

Different previous reviews [9,24] have shown some interventions to effectively reduce text anxiety in this population, including hypnotherapy, aromatherapy, music therapy, and test-taking education. However, relaxation training, including diaphragmatic breathing, progressive muscle relaxation, and biofeedback-assisted relaxation training, found no difference in anxiety scores post-intervention, despite the participants reporting the training as useful [25]. This is in line with the results of our study. No differences were detected in either of the two relaxation techniques used compared to the control group. However, some studies combine relaxation techniques with guided imagery, improving test and grade point average scores [26,27]. Therefore, it appears interesting to combine relaxation techniques with other therapies, like guided imagery or others, that have already been proven effective to add their benefits.

Regarding the effect of anxiety on academic performance, our study showed that students with moderate levels of anxiety performed better than those with high levels of anxiety, although the authors Dawood et al. [7] did not find this association, which could be due to the different sociodemographic characteristics of their students. If we consider the student’s academic level, authors such as Vaz et al. [23] found a significant negative relationship between test anxiety scores and academic achievement in undergraduate nursing students compared with higher academic undergraduate nursing students who experienced less test anxiety.

Considering the sociodemographic factors that may affect observed associations, age did not correlate with the level of anxiety in our study, as occurred in previous research, such as that carried out by Dawood et al. [7]. One strength of our study is the analysis of sex differences on anxiety levels, and we found a higher lack of confidence among female students. This is an important point to consider as confidence should be more reinforced among women who, nowadays, are still the majority of nursing students.

Qualitative studies [6,28] have shown that nursing students perceive that teachers provide essential support during exams. Teachers can help their students by accommodating the test environment and facilitating relaxation techniques, as it provides an added level of fairness in the learning evaluation process and could decrease anxiety associated with exams. Other factors that have been shown to influence examination anxiety among undergraduate nursing students are the learning process, perceptions related to examination, learning patterns, and over-expectations related to learning outcomes [23]. It makes us realize it is beneficial to work on reducing exam anxiety from the onset of a nursing degree, as different factors could modulate this anxiety. Nurse educators can significantly impact student outcomes by recognizing test anxiety, intervening early, and implementing effective, supportive strategies [24]. In the case of our study, we found that providing social support is associated with less concern and interference with external factors, better confidence, and emotionality, as explained by Dundas et al. [14], who also achieved favorable results in pre-test anxiety in terms of self-concept and self-acceptance after close contact. The decrease in anxiety that accompanies this acceptance can also, paradoxically, allow them easier access to the knowledge they possess at the time of the exam. This same study indicated that relaxation through direct contact and mindfulness-guided breathing could reduce dysfunctional perfectionist tendencies in college students [14]. Phang et al. [15] and Ratanasiripong et al. [16] also found that the university population in health sciences exposed to mindfulness sessions with direct contact and empathy by the faculty before the exam showed lower anxiety levels. However, Ratanasiripong et al. [15], who considered the stress level measured with the Perceived Stress Scale, did not find the same result after administering this last relaxation technique.

Educators must broaden their knowledge base regarding the experience of test anxiety while facilitating the learner’s progression, whether in the nursing education program or the facility’s orientation program. Techniques such as guided imagery, diaphragmatic breathing, and muscle relaxation improve the clarity of mind and, ultimately, lead to positive outcomes. The study by Cho et al. [13] found that guided breathing led to a sense of mastery over thoughts and emotions and the feeling of being able to perceive them as transient mental events rather than identifying with them or believing that thoughts and emotions are reflections of reality. Therefore, the decentering caused by a greater concentration on the breath helps to disengage from self-criticism, rumination, and anxiety that can arise when reacting to negative thought patterns [13]. Preparing learners in terms of note-taking in the classroom and strategies that improve concentration and comprehension when reading the text is an essential component of test preparation for successful outcomes. The learner should be encouraged to take practice examinations and read the rationales for the correct/incorrect answers to understand individual strengths and weaknesses in a specific course matter [6]. All these are important to get a high self-concept and intrinsic motivation among students as these are significant mediators in the relationship between self-concept and academic achievement [29].

Finally, we would like to note that despite the strengths of this study, some limitations can be found. It was impossible to design an experimental study due to the added nervousness it would cause the students participating in a randomized clinical trial with a pre-test and post-test, which would also lead to an excessively long test. In addition, this study was carried out with a small convenience sample in a single center. Some sociodemographic variables, such as socioeconomic status, number of tests already done in the past, preceding experiences of relaxation techniques, etc., were not analyzed due to limited time for measurements and interventions. Furthermore, the generalizability of our results is limited by the fact that three-quarters of the sample were female, but this was unavoidable because it is already known that the majority of student nurses are still female due to traditional gender roles. Nevertheless, the assignment of students to the groups was randomized, which would ensure the comparability of the groups by eliminating the confounding factor of gender. Therefore, we recommend carrying out multicenter studies in an experimental way overpassing these limitations and trying different interventions or combining them to improve the knowledge of this phenomenon and the recommendations for nurse educators. In addition, future research might also measure physiological variables before and after the test, as well as psychological disorders and substance use that affects activation levels, to enhance the objectivity and validity of study results, as this study measured anxiety with a self-reported questionnaire. Several measurements of anxiety levels could also be made by applying these techniques from the beginning of the courses to study the long-term effects of the techniques, as our study found no short-term effects.

## 5. Conclusions

Test anxiety is high-moderate among undergraduate nursing students. Moderate levels of anxiety have been shown to improve academic performance. Therefore, there is a necessity to develop effective interventions to decrease this examination anxiety. Our study did not find that guided breathing or social support interventions were effective for this purpose. However, combining these relaxation techniques with others shown to be effective could reinforce their positive effect. More studies with larger samples are necessary to study this phenomenon in more depth and to be able to give better recommendations to nursing educators. Nonetheless, starting to work on this anxiety from the beginning of the nursing courses appears to be a good strategy, striving to improve the students’ confidence in their abilities.

## Figures and Tables

**Table 1 healthcare-11-00574-t001:** Characteristics of the participants.

Variable	Category	Frequency (%)	Mean (SD)	Item Mean
Sex	Male	27 (22.7)	-	-
	Female	92 (77.3)	-	-
Age	-	-	22.7 (4.62)	-
Knowledge score	-	-	8.45 (1.26)	-
Intervention group	Control	41 (34.5)	-	-
	Social support	37 (31)	-	-
	Guided breathing	41 (34.5)	-	-
Anxiety scale	Emotionality	-	22.82 (6.05) Range 8–32	2.85 (0.96)
	Worry	-	29.55 (5.08) Range 10–40	3.29 (0.8)
	Lack of confidence	-	16.39 (4.25) Range 6–24	2.68 (0.75)
	Interference	-	11.67 (3.64) Range 5–20	2.33 (0.93)
	Total	-	80.4 (10.7)	2.87 (0–88)

**Table 2 healthcare-11-00574-t002:** Bivariate analysis of the anxiety scale.

Total Anxiety Scale Score			
Variable	Category	Score (SD)	Contrast
Sex	Male	79.59 (10.61)	T = −0.459; *p* = 0.648
	Female	80.66 (10.79)	
Age			R = 0.173; *p* = 0.060
Knowledge scale score			Rho = −0.222; *p* = 0.015 *
Intervention group	Control	81 (9.49)	F = 0.803; *p* = 0.451
	Contact	78.6 (11.1)	
	Guide breathing	81.5 (11.5)	
**Subscale: Emotionality**			
Variable	Category	Score (SD)	Contrast
Sex	Male	21.9 (5.98)	T = −0.867; *p* = 0.388
	Female	23.1 (6.08)	
Age			R = 0.152; *p* = 0.098
Knowledge scale score			Rho = −0.272; *p* = 0.003 **
Intervention group	Control	22.7 (5.59)	F = 0.764; *p* = 0.468
	Contact	22 (6.32)	
	Guide breathing	23.7 (6.28)	
**Subscale: Worry**			
Variable	Category	Score (SD)	Contrast
Sex	Male	28.4 (5.12)	Z = −1.469; *p* = 0.142
	Female	29.9 (4.95)	
Age			Rho = 0.125; *p* = 0.176
Knowledge scale score			Rho = −0.183; *p* = 0.047 *
Intervention group	Control	30.1 (4.76)	Χ^2^ = 2.747; *p* = 0.253
	Contact	28.4 (5.28)	
	Guide breathing	30 (4.96)	
**Subscale: Lack of confidence**			
Variable	Category	Score (SD)	Contrast
Sex	Male	18.26 (3.61)	Z = −2.532; *p* = 0.011 *
	Female	15.85 (4.29)	
Age			Rho = −0.025; *p* = 0.789
Knowledge scale score			Rho = 0.314; *p* = 0.001 **
Intervention group	Control	17.1 (3.88)	Χ^2^ = 1.887; *p* = 0.389
	Contact	16.5 (4.49)	
	Guide breathing	15.6 (4.37)	
**Subscale: Interference**			
Variable	Category	Score (SD)	Contrast
Sex	Male	11 (3.60)	Z = −1.084; *p* = 0.278
	Female	11.9 (3.65)	
Age			Rho = 0.112; *p* = 0.224
Knowledge scale score			Rho = −0.358; *p* = 0.001 **
Intervention group	Control	11.1 (3.65)	Χ^2^ = 2.050; *p* = 0.359
	Contact	11.7(3.63)	
	Guide breathing	12.2 (3.65)	

F: Fisher’s F; R: R de Pearson; Rho: Rho de Spearman; T: Student’s T; X^2^: Kruskal Wallis test; Z: Mann Whitney test. * *p* < 0.05; ** *p* < 0.01.

## Data Availability

All data generated or analyzed during this study are included in this published article [Appendix A].

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
