# Peer review of "Effectiveness of Guided Breathing and Social Support for the Reduction of Pre-Exam Anxiety in University Students: A Factorial Study"

_healthcare, 2023, doi:10.3390/healthcare11040574_

Round 1

Reviewer 1 Report (Previous Reviewer 1)

I see consderable improvement of the paper in this second version. I advise to replace misspelled Sperman by 'Spearman's Rho'

Author Response

We have modified the word according to your indication.
Thank you very much for your review.

Reviewer 2 Report (Previous Reviewer 4)

See comments in the attached file

Author Response

1) At Line 144 gender academic year any analyses, so I wonder why it was collected.

We collected data on the variable academic year in case there were any students from another year, but in the end all the students were from the same year, so this variable was not studied in the analyses. Following your question, we considered it better to eliminate this variable from the socio-demographic data as it does not provide relevant information.

2) Section 2.4 (Statistical Analysis) has surely improved, but there is something to rearrange or clarify.

The section 2.4 has been modified following the recommendations.

3) It is not clear why now Table 1 has two columns with the same heading (Mean): this is confusing for the reader!

The table 1 was modified changing the fifth column name by “Item mean”.

4) Table 2 has improved, but there is still something to check and rearrange.

Table 2 was modified following the recommendations.

Thank you for your review.

This manuscript is a resubmission of an earlier submission. The following is a list of the peer review reports and author responses from that submission.

Round 1

Reviewer 1 Report

This study concerning two relaxation techniques for nurse students before entering an exam is set up as a factorial design. However, I doubt whether the correct types of analyses are performed. One would expect some F-tests used for this design. Furthermore, since no effect is found of the relaxation trainings, the interpretation of results should be much more cautious than is the case in this paper. Furthermore, the paper shows several weaknesses: in English language, in lack of clarity with respect to the description of the interventions, sampling precision, analysis techniques and sampling method. On the whole, the results are not convincing, due to design and language flaws. Moreover, the interpretation of the results is not logical: why set up a study with relaxation techniques of which it was known beforehand that results were  meager? The authors refer to such a study (Prator, 2013) themselves. On the whole, this paper is not strong enough – neither in design, nor in careful writing - for publication in this journal.  

Reviewer 2 Report

Dear Authors,

I'm thankful to review the paper entitled „Effectiveness of guided breathing and social support for the reduction of pre-exam anxiety in university students: A factorial study”. The issue raised by the authors is very important and necessary in public discussion. Nevertheless, in my opinion, the paper needs minor changes before can be published in Healthcare. I think the topic is certainly worth studying, and I strongly encourage the authors to make some changes, especially in the discussion section.

In the Sample section - there is no characteristic or description of the study group. All we know is that the average age is 22.7 and SD = 4.6.

Page 4, Line - 187 There is a sentence: The worry subscale showed the highest score before the assessment test. - Is it objective? Are the authors entitled to draw such conclusions? After all, this scale had the most questions (10) compared to the others, so it is obvious that it will always have the highest average (I assume that since there are 10 items, the respondents could get between 10 and 40 points, and the average was 29.55, and in the case of e.g. Lack of confidence range is between 6 and 24 points, therefore the average scores are lower (M = 16.39) In my opinion, in Table 1 there should be a range of points for each of the subscales.

Page 7, Line 238 Authors wrote: One strength of our study is the analysis of the influence of gender on anxiety level, and we found a higher lack of confidence among female students.

It is difficult to agree with this opinion because the authors did not study the influence of gender. What statistical analyzes have been performed to be authorized to draw such conclusions? Ordinary correlations or differences analyzes do not allow for such conclusions.

Also in the discussion, the authors argue that relaxation helps, although the results of these studies do not confirm this. This is also a hastily drawn conclusion, it is not proven in these analyses. (In conclusion, we find a sentence in line 303: Our study did not find that guided breathing or social support interventions were effective for this purpose.)

A similar inaccuracy appears on line 230: Regarding the effect of anxiety on academic performance, our study showed that the students with lower anxiety performed better than those with high anxiety levels.

The authors wrote that students with a low level of anxiety - probably should be moderate anxiety instead lower because only 1 student had a low level of anxiety.

Page 7, Line 252 Authors wrote: In the case of our study, we found that providing social support is related to less concern and interference with external factors, better self-knowledge and emotionality.

What external factors?? better self-knowledge  - The authors did not check this variable, they checked the lack of confidence.

Why in table 2, Score is in %, how to understand it?, is it M and SD? what is in brackets?

Page 4, Line 186 - the authors wrote mean of anxiety… and in the table is Median.  Is it the mean or the median?

In limitation, the authors should take into account that more than 3/4 of the respondents were women. Maybe also in the description of the groups, it is worth describing how the number of respondents was distributed, taking into account gender in particular groups. (Was there a fairly identical distribution of men in all groups?).

Sincerely

Reviewer

Reviewer 3 Report

Thank yoyu for the oportunity to review the article entitled ”Effectiveness of guided breathing and social support for the reduction of pre-exam anxiety in university students: A factorial 3 study”

The manuscript is, in general, well-written (the form is nice), but there are some issues that must be changed, corrected or replaced (the content must be corrected).

Here are some of my comments:

-        Please revise the paragraphs (lines 33-36) in order to be clear for the Reader. It is not clear the explanation and it is a combination of ideas that must by re-written and adjusted to the goal and variables of the study. The Introductions ection must clarify the concepts used in the research - ” This situation presents differential characteristics to the general feeling of anxiety, with a phobic quality that is superior to other anxiety disorders [3]. Associated with this feeling of anxiety, there is a situation perceived as a danger resulting in the sympathetic nervous system's response through the activation of body systems for the optimal response to it”

-        Where were the participants included in the researcher evaluated before? It is well-known that some types of personalities are less prone to relax, and some others are more prone to manifest anxiety and stress.

-        Some of the relaxation techniques need to be taught – meaning that the participant need time to practice that type of relaxation. Please describe the methodology from this perspective.

-        Lines 47-48: ”When considering these new coping and control strategies” – which ones?

-        Mixing ideas can be found also in lines 96-102:  ”this study aimed to determine the effect of guided breathing and social support on test anxiety among nursing students administered minutes before a final  knowledge assessment test and its association with the sex and age of the participants. As a research question we posed whether guided breathing or social support is effective in reducing anxiety and it performance at evaluation test, comparing anxiety levels after these two techniques with no intervention. The choice of these specific techniques is due  to prior knowledge of them by the students, not excessively distracting them, apart from not altering the time available to take the test” – The authors considered that sex and age are the most important variables? Also, the authors presented into the Introductions section about Jacobson method, but this method was not applied for the present research.

-        So, the Introduction must be strongly related to the content of the manuscript and to sustain the aims of the study.

-        Line 112-113: the authors refer to ” All participants were asked about their health and medication intake that may affect their stress level” - but their goal was to reduce anxiety.

-        Line 113-114: ”The participants who did not take any substance were selected for the research.” – the authors previously sustained that they questioned medication that may affect stress. How do they evaluate that those medicines were reducing/increasing the level of stress? After that, the authors sustained that they eliminated all participants that were proved to take medicines.

-        Male/female number of participants is unbalanced – results cannot be generalizable.

-        Please insert clearly inclusion/exclusion criteria

-        Please mention the period of time the study was developed, the number of previous meetings, if participants were rewarded, etc

-        There are a lot of English spelling mistakes – please correct them (EG: imagery (line 126), Rgarding(line 230), etc etc

Reviewer 4 Report

In general, the study presented in this paper seems interesting and well conducted.

In particular, regarding the statistical analysis, I think that the authors have used a correct methodology, both in the selection of the sample together with the evaluation of the statistical power (presented in section 2.2) and in the analyses described in section 2.4.

But I noticed some tiny bit of confusion or inconsistency in the presentation of those methods or in the results.

More in details:

1)   I would suggest expanding section 2.1, adding some more explanations on factorial study design.

2)   Line 141: “Sociodemographic data: Composed of three items, age, sex, and gender”; the items sex and gender are not the same?

3)   At lines 165-166 it is stated that “the normality of the knowledge level variable was tested with the Kolmogorov-Smirnov test”; but then “a p > 0.05 value was obtained for Anxiety score”; so it is supposed that not only knowledge level variable was tested for normality, but also Anxiety score; please clarify this issue rearranging the text of section 2.4 accordingly.

4)   The data presented in Table 2 are somehow confused and not completely consistent with what was stated in Materials and Methods section; in particular, for the contrast analysis, why for the same variable you used sometimes the indicator T and sometimes Z? Or sometimes F and sometimes Χ2? Or sometimes R and sometimes Rho? Please check and if it is the case correct, or add a more detailed description in section 2.4, and possibly some footnotes to Table 2, in order to describe the indicators used.

5)   Have you considered other possible variables that could influence the anxiety level, such as socioeconomical status, number of tests already done in the past, preceding experiences of relaxation technique, etc.?

6)   In general, the text should be checked for English language and for typos; for example: “imagary” (lines 226 and 228), “Rgarding” (line 230), “themor” (line 275).